# Determinants of Health Facility Utilization at Birth in South Sudan

**DOI:** 10.3390/ijerph16132445

**Published:** 2019-07-09

**Authors:** Justin Bruno Tongun, David Mukunya, Thorkild Tylleskar, Mohamedi Boy Sebit, James K Tumwine, Grace Ndeezi

**Affiliations:** 1Centre for International Health, University of Bergen, 7804 Bergen, Norway; 2Department of Paediatrics, University of Juba, Juba P.O. Box 82, South Sudan; 3Department of Internal Medicine, University of Juba, Juba P.O. Box 82, South Sudan; 4Department of Paediatrics and Child Health, School of Medicine, College of Health Sciences, Makerere University, Kampala P.O. Box 7072, Uganda

**Keywords:** childbirth, health-facility-births, skilled birth attendant, reproductive health, South-Sudan

## Abstract

South Sudan has a high maternal mortality ratio estimated at 800 deaths per 100,000 live births. Birth in health facilities with skilled attendants can lower this mortality. In this cross-sectional study, we determined the level and determinants of health facility utilization and skilled birth attendance in Jubek State, South Sudan. Mothers of children aged less than two years were interviewed in their homes. Multivariable regression analysis was performed to determine factors associated with health facility births. Only a quarter of the mothers had given birth at health facilities, 209/810 (25.8%; 95% CI 18.2–35.3) and 207/810 had a skilled birth attendant (defined as either nurse, midwife, clinical officer, or doctor). Factors positively associated with health facility births were four or more antenatal visits (adjusted odds ratio (AOR) 19; 95% CI 6.2, 61), secondary or higher education (AOR 7.9; 95% CI 3, 21), high socio-economic status (AOR 4.5; 95% CI 2.2, 9.4), and being primipara (AOR 2.9; 95% CI 1.5, 5.4). These findings highlight the need for efforts to increase health facility births in South Sudan.

## 1. Introduction

The global maternal mortality ratio (MMR) has dropped by 44% in the last 25 years [1]. This decline varies widely between low- and high-income countries. Low-income countries contribute 99% of the maternal deaths in the world, and sub-Saharan Africa accounts for 66% of these deaths [1].

Often, maternal deaths are due to direct obstetric causes such as postpartum haemorrhage, obstructed labour, sepsis, unsafe abortion, and hypertension [2]. These complications can be mitigated by encouraging mothers to give birth at health facilities with the help of skilled birth attendants [3]. Mothers who give birth at health facilities are also less likely to die or lose their new-borns [4,5,6]. Reduction of mortality in health facility births is, largely, due to skilled health workers’ ability to prevent, treat, or control fatal obstetric and neonatal complications.

South Sudan has one of the highest MMR and neonatal mortality rates (NMR) in the world. The MMR was estimated to be 800 per 100,000 live births in 2015, while the NMR was about 39 per 1000 live births in the same year [7,8]. These figures are probably higher in rural areas and those involved in the civil war [9]. The South Sudan Demographic Household survey in 2010 report showed that only 12% of mothers used health facilities during childbirth with skilled birth attendants [10]. For South Sudan to meet the targets of the third Sustainable Development Goal (SDG) of reducing the MMR to less than 70 per 100,000 live births, and the neonatal mortality rate to less than 12/1000 live births [11] interventions that reduce both maternal and neonatal mortality and morbidity must be implemented [12,13]. One such intervention is scaling up health facility births and skilled birth attendance. To design interventions that promote health facility births in South Sudan, up to date context-specific data are needed [14].

We report the level and determinants of health facility utilization and skilled birth attendance during childbirth in Jubek State, South Sudan.

## 2. Subjects and Methods

### 2.1. Study Design

This was a cross-sectional study carried out among mothers of children aged 0–23 months.

#### 2.1.1. Setting

This study was carried out from October 2016–December 2016 in four counties in Jubek State, South Sudan. Jubek State has 12 Counties and one city council (Juba) with a population of about 500,000 [15]. Most of the inhabitants are ethnic *Bari* who speaks the *Bari* language. Farming is the main economic activity. The survey was carried out in four rural counties of Lodu, Luri, Mangala, and Rajaf [15].

#### 2.1.2. Sampling

We used a two-stage sampling method, which is; basically, a modification of the World Health Organization (WHO) expanded programme on immunization (EPI) method for estimating vaccination coverage [16]. We listed all 43 villages and corresponding populations in the four counties and selected 30 out of 43 the villages by probability proportionate to size [16]. In each village, we selected an index house randomly and proceeded to the nearest house. The next house was selected by picking the house nearest to the index household; the one whose door was closest. From each household, we recruited one mother-infant pair. This process was repeated until a total of 27 mother-infant pairs had been interviewed from each of the 30 villages, giving us a total of 810 participants. The details of the sampling procedure have been published in the Global Health Action Journal [17].

Eight trained research assistants conversant with the study area and fluent in *Bari* (the local language) collected the data. The interviews were done in a private area in the mother’s home, away from the other members of the family.

### 2.2. Study Participants

We included mothers of children aged 0–23 months; if a mother had two children born in the last two years, only the youngest was selected. We excluded mother-infant pairs who were not residents in the village, children with no mothers, and those whose mothers were not mentally sound to complete the interview.

### 2.3. Variables

The outcome variables included: the place of birth and skilled birth attendance. Place of birth categorized as a healthcare facility or other (home, on the way to a health facility, traditional birth attendant’s house, etc.). Mothers were also asked about the person who assisted them during childbirth, and this was categorized as skilled birth attendance if they reported a healthcare worker (defined as either nurse, midwife, clinical officer, or doctor) or unskilled birth attendance. Other variables included the mothers’ age categorised as ≤19, 20–24, 25–29, and ≥30 years; marital status classified as single or married; mothers’ education categorized as no formal education, primary, ≥secondary; antenatal care visits categorised as none, 1 to 3, ≥4; parity classified as 1 or >1 and socio-economic status categorized as quintiles Q1 (poorest), Q2 (poor), Q3 (medium), Q4 (less poor) and Q5 (least poor). The socioeconomic status was calculated using multiple correspondence analysis [18] base on: (a) ownership of assets such as car, phone, radio, television, fridge, cupboard, bicycle, motorcycle, house, land, (b) fuel use for cooking, and (c) assessment of household dwelling characteristics like material of the floor, roof, and house type.

### 2.4. Data Analysis

We present continuous variables as means and standard deviations; and categorical variables as proportions. We used logistic regression to assess factors associated with health facility birth and skilled birth attendance. 

Factors associated with health facility births from the literature and those with a *p*-value ≤ 0.25, and not strongly collinear with other independent variables were entered in the initial multivariate logistic regression model. We assessed for collinearity and considered factors with a variance inflation factor more than ten strongly collinear. In case of collinearity, the factor with a stronger measure of association with the dependent variable was retained, and the other dropped from the model. We used STATA version 14 (STATA Corp LLC, Texas, TX, USA) with survey set command adjusting for the multistage sampling in the data analysis.

### 2.5. Ethics

We obtained ethical approval from the Directorate of Planning, Budgeting, and Research in the Ministry of Health in South Sudan—reference number SMOH/E/JS/44.K.1. Official letters of permission were presented to the county commissioners and village chiefs. Written informed consent was obtained from the study participants after providing information about the purpose and procedures of the study. We obtained a thumbprint from participants who were unable to write. Privacy and confidentially measures were maintained throughout the study. No compensation was given to the study participants. We addressed the participants’ questions regarding the survey accordingly.

## 3. Results

### 3.1. Socio-Demographic and Birth Characteristics

A total of 810 mothers were included in this survey, Table 1. The mean and standard deviation (SD) of the age of mothers was 26.6 (5.5) years. Most of the mothers were married; over half had no formal education. Only a quarter of the mothers gave birth at a health facility, 209/810 (25.8%; 95% CI 18.2–35.3). Of the 810 mothers, 204 had had a skilled birth attendant, and five did not. Another three mothers who did not give birth at a health centre said they had a skilled birth attendant. 

### 3.2. Factor Associated with Health Facility Birth and Skilled Birth Attendance at Birth

Factors linked to health facility births at bivariable level included mother’s age, mother’s education, one or more antenatal care visits, high socio-economic status, and maternal parity greater than one, Table 2.

Factors positively associated with health facility births in the multivariable analysis included, mother’s education status, antenatal care visits, socio-economic status, and parity, Table 2.

Factors associated with skilled birth attendance in the multivariable analysis included mothers’ education status, antenatal care visits, and socio-economic status, Table 3.

## 4. Discussion

This survey found low levels of utilization of health facilities and skilled birth attendance during childbirth; only a quarter of mothers utilized health facilities or received skilled birth attendance during childbirth.

The proportion of mothers who gave birth in a health facility was higher than that observed in the nation-wide South Sudan Household Survey (SSHS) conducted in 2010, which showed that only 12% of mothers gave birth in a health facility [10]. These proportions cannot be compared as the geographical coverage was different. In fact, the study area surveyed is close to Juba. In the rest of the country, health facility utilization is likely to be lower than what we observed. Therefore, the proportion of mothers who give birth at health facilities is still alarmingly deficient. This could be partly due to the insecurity [19] resulting from the on-going civil unrest in South Sudan, which has hindered improvement in the necessary infrastructure. Instability also discourages mothers from utilizing services, especially during the night [19,20,21]. Furthermore, socioeconomic consequences and shocks of war lead to poor health service delivery and utilization [9,22,23]. Also, the low use of health facilities for childbirth could be due to supply-side reasons such as poor quality of health care [24]. Low-quality health care is known to discourage women from using health facilities [21]. There could also be socio-cultural factors such as fear of dignity violation [25] that discourage mothers from seeking health care during childbirth.

The factors associated with giving birth at a health facility or receiving skilled birth attendance during childbirth included: having attended more antenatal care (ANC) visits, mother’s education status, higher socio-economic status, and being a first-time mother. These factors were comparable to the results of a recent study on the risk factors for not using health facility at birth in South Sudan [26]. 

Antenatal care visit was associated with health facility births. Mothers who had attended at least four ANC visits were 19 times more likely to deliver in health facilities, compared to others who did not attend any antenatal care. This was similar to findings from Tanzania [27]. Women who attended ANC several times could have gained knowledge and understanding of the advantages of a facility birth. Further, women could also have become familiar with the health workers during the ANC visits and were inspired to give birth in health facilities [28].

We also found that the higher the mothers’ education, the more likely she was to give birth in a health facility. These mothers were more likely to engage in health seeking behaviours due to higher knowledge levels, and indirectly through education’s influence on other factors such as income [29,30]. 

In our study, mothers of higher socioeconomic status were more likely to give birth at a health facility or receive skilled assistance when giving birth. Women of lower economic status might have had difficulties in finding transport or meeting indirect costs related to childbirth, and hence barred from seeking institutional delivery [19].

A recent qualitative study by Wilunda and colleagues found political instability/ inter-communal conflict, lack of health facility infrastructure, shortage of medical supplies, socio-cultural practices, perception of childbirth, quality of obstetric care as potential factors influencing health facility-based birth in South Sudan [19]. In sub-Sahara Africa, a body of evidence found several factors that directly influenced health facility utilization during childbirth [31,32,33]. These factors include poor transport infrastructure, lack of finance for transport, indirect cost of obstetric care, distant to health facility, shortage of skilled health workers, suboptimal training of health workers, poor quality of obstetric care characterised by long waiting time, weak referral system and poor staff interpersonal skills, and attitude.

The findings in this study highlight the need to promote health facility births, antenatal care, the general economic situation of the population, and girl child education. The government in South Sudan could emulate evidence-based integrated multi-level practices that have worked in other settings such as: introduction of policies and programmes that support health facility-based birth, training of health workers, building more health facilities, improving referral systems, and addressing issues related to affordability and financial risk associated with access to obstetric services [13,31]. Furthermore, the stakeholders should pay close attention to traditional, socio-cultural and socio-economic factors that are critical in delayed decision making in seeking obstetric services, and address women’s concerns regarding supportive attendance during birth [34].

One strength of this study is that we conducted a community-based survey, in a period of insecurity, where most studies on the subject are either hospital-based or qualitative. To the best of our knowledge, this study provides the first community-based estimate of health facility births in South Sudan since 2010. These data can be used to generate better estimates of health facility births and skilled childbirth in rural South Sudan. However, we did not ask for a history of pregnancy, measure some factors such as distance to the health facility, and complications during previous childbirth, which are essential determinants of place of birth. Lastly, this study was conducted in one state and is unlikely to be generalizable to the whole country.

## 5. Conclusions

We found that only a quarter of the women gave birth in a health facility in an area close to the capital Juba in South Sudan. Other areas of the country are likely to have even lower attendance. Factors positively associated with health facility births were antenatal care visits, secondary or higher maternal education, high socio-economic status, and primiparity. These findings highlight the need for efforts to increase health facility births and skilled attendance in South Sudan. There is also an urgent need to conduct a broader community-based mixed methods study to provide an in-depth understanding of barriers and facilitators of health facility utilization during childbirth in South Sudan.

## Figures and Tables

**Table 1 ijerph-16-02445-t001:** Baseline characteristics of mothers in a community survey in Jubek State, South Sudan.

Characteristics	All Participants	Health Facility Births
	*N* = 810	*N* = 209 (25.8%)
	*n* (%)	*n* (%)
Age of the mother		
≤19	89 (11.0)	35 (16.8)
20–24	195 (24.1)	67 (32.1)
25–29	279 (34.4)	59 (28.2)
30–34	173 (21.4)	40 (19.1)
≥35	74 (9.1)	8 (3.8)
Marital status		
Single	17 (2.1)	6 (2.9)
Married	793 (97.9)	203 (97.1)
Mother’s education		
None	516 (63.7)	63 (30.1)
Primary	228 (28.2)	96 (45.9)
≥Secondary	66 (8.2)	50 (23.9)
Mother’s employment		
Unemployed	690 (85.2)	166 (79.4)
Employed	120 (14.8)	43 (20.6)
Antenatal care visits		
None	165 (20.4)	4 (1.9)
1–3	444 (54.8)	98 (46.9)
≥4	201 (24.8)	107 (51.2)
Parity		
1	138 (17.0)	61 (29.2)
>1	672 (83.0)	148 (70.8)
Socio-economic quintiles		
Poorest (Q1)	286 (35.3)	28 (13.4)
Poor (Q2)	95 (11.7)	20 (9.6)
Medium (Q3)	114 (14.1)	17 (8.1)
Less poor (Q4)	154 (19.0)	51 (24.4)
Least Poor (Q5)	161 (19.9)	93 (44.4)

**Table 2 ijerph-16-02445-t002:** Bivariate and multivariate logistic regression analysis of health facility birth in a community survey in Jubek State, South Sudan

Characteristic	Bi-Variable	Multivariable Model l
	*N* = 810	*N* = 810
	OR (95%CI)	AOR (95%CI)
Mother’s age		
≤19	1	1
20–24	0.8 (0.46, 1.4)	0.9 (0.44, 1.9)
25–29	0.4 (0.24, 0.7)	0.8 (0.31, 2.02)
30–34	0.5 (0.2, 1.1)	0.9 (0.32, 2.7)
≥35	0.2 (0.06, 0.6)	0.5 (0.14, 1.9)
Marital status		
Married	1	
Single	1.6 (0.46, 5.4)	-
Mother’s education		
No formal education	1	1
Primary	5.2 (3.2, 8.5)	3.1 (1.9, 5.2)
≥Secondary	22 (11, 46)	7.9 (3, 21)
Mother’s employment		
Unemployed	1	1
Employed	1.8 (0.94, 3.3)	1.2 (0.6, 2.4)
Antenatal care visits		
None	1	1
1–3	11 (4.2, 31)	5.2 (1.7, 16)
≥4	46 (15, 140)	19 (6.2, 61)
Parity		
1	2.9 (1.8, 4.5)	2.9 (1.5, 5.4)
>1	1	1
Socio-economic quintiles		
Poorest (Q1)	1	1
Poor (Q2)	2.5 (1.2, 4.9)	1.7 (0.8, 3.6)
Medium (Q3)	1.6 (0.64, 4.1)	1.3 (0.5, 3.0)
Less poor (Q4)	4.6 (2.3, 9.3)	2.4 (1.1, 5.0)
Least poor (Q5)	12 (7.0, 24)	4.5 (2.2, 9.4)

**Table 3 ijerph-16-02445-t003:** Bi-variable and multivariable logistic regression analysis of skilled birth attendance in a community survey in Jubek State, South Sudan

Characteristic	Bi-Variable	Multivariable Model 2
	*N* = 810	*N* = 810
	OR (95%CI)	AOR (95%CI)
Mother’s age		
≤19	1	1
20–24	0.93 (0.54, 1.60)	1.18 (0.55, 2.52)
25–29	0.44 (0.25, 0.75)	0.88 (0.37, 2.12)
30–34	0.51 (0.22, 1.18)	1.10 (0.40, 3.08)
≥35	0.21 (0.07, 0.61)	0.61 (0.17, 2.14)
Marital status		
Single	1	
Married	0.48 (0.18, 1.3)	-
Mother education		
No formal education	1	1
Primary	5.2 (3.1, 8.7)	3.1 (1.79, 5.37)
≥Secondary	22.9 (11.4, 45.9)	8.2 (3.18, 21.27)
Mother employment		
Unemployed	1	1
Employed	1.6 (0.89, 3.1)	1.05 (0.53, 2.08)
Antenatal care visits		
None	1	1
1–3	8.7 (3.5, 21.9)	3.93 (1.52, 10.15)
≥4	36.4 (11.9, 111.7)	15.17 (5.53, 41.58)
Parity		
>1	1	1
1	2.7 (1.8, 4.3)	2.9 (1.65, 5.16)
Wealth Quintiles		
Poorest (Q1)	1	1
Poor (Q2)	2.5 (1.3, 4.8)	0.61 (0.80, 3.54)
Medium (Q3)	1.6 (0.67, 3.9)	1.26 (0.55, 2.85)
Less poor (Q4)	4.3 (2.2, 8.3)	2.20 (1.123, 4.30)
Wealthiest (Q5)	12.6 (6.8, 23.2)	4.64 (2.38, 9.03)

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
