# Peer review of "Determinants of Health Facility Utilization at Birth in South Sudan"

_ijerph, 2019, doi:10.3390/ijerph16132445_

Round 1
Reviewer 1 Report
Abstract: Very clear summary of the findings of the article.
Introduction: The author described the MMR and NMR in South Sudan. I would recommend adding the reporting years.
How about the earlier data of health facility utilization at birth in South Sudan?
For line 47, one reference should be added. Reader would appreciate it if the authors could introduce more about the background information.
The authors clearly state the purpose and objective of the present study.
Subjects and Methods
How sample were recruited?
Inclusion criteria should be listed.
Results:
How about the pregnancy history in this population?
The definition of socio-economic quintiles is not clear. Household annual income? The currency and equal to USD.
Of the total sample, how many mothers had a skilled birth attendance?
In 3.2. I would recommend the author added the results of factors associated with skilled birth attendance.
Discussion:
The author provides context for the results and compares the findings of this study to other area and other population. Besides mother’s education and socioeconomics status, are they any other potential associated factors? Give some suggestions for next steps that the Sudan government could take to increase access to health facility utilization. It could be discussed more.
The authors might consider adding some text describing the next steps for research.
Author Response
We thank you for your detailed and professional suggestions and comments. We are grateful and know that they will make our manuscript professional and easier to read. We have revised the manuscript according to the comments. We have made point by point responses as follows:
ABSTRACT
Comment:
Very clear summary of the findings of the article.
Response:
Thank you for this kind comment
INTRODUCTION
Comment:
The author described the MMR and NMR in South Sudan. I would recommend adding the reporting years.
Response:
We have added the reporting year for both MMR and NMR. Line 42- 43; page 1
Comment:
How about the earlier data of health facility utilization at birth in South Sudan?
Response:
We have included earlier data of health facility utilization at childbirth. Line 44-45; page 1
Comment:
For line 47, one reference should be added. Reader would appreciate it if the authors could introduce more about the background information.
Response:
We have added two references. Line 49; page 2
Comment:
The authors clearly state the purpose and objective of the present study.
Response:
Thank you very much for the comment
SUBJECTS AND METHODS
Comment:
How sample were recruited?
Response:
In brief, we used a two-stage sampling method. First, we selected 30 out of the 43 villages by probability proportionate to size. In each village we selected an index household randomly and proceeded to the nearest house. We recruited one mother-infant pair from each household. A total of 27 participants were recruited from each of 30 villages giving us 810 participants. Line 69-73; page 2.
Comment:
Inclusion criteria should be listed.
Response:
We have clarified the inclusion and exclusion criteria and it reads as follows; we included mothers of children aged 0-23 months. If a mother has two children born in the last two years, only the youngest child was selected. Excluded mother-infant pairs who were not residents in the village. Excluded children with no mothers and mothers who were not mentally sound to complete the interview. Line 78-81; page 2.
RESULTS
Comment:
How about the pregnancy history in this population?
Response:
We did not obtained information about pregnancy history in this population and we acknowledged this is a limitation. Line 203-204; page 7.
Comment:
The definition of socio-economic quintiles is not clear. Household annual income? The currency and equal to USD.
Response:
We did not estimate average household income. However, we have clarified about the socio-economic quintiles and it reads as follows: Socioeconomic status was calculated using multiple correspondence analysis (Howe et al. 2008) base on: a) ownership of assets such as car, phone, radio, television, fridge, cupboard, bicycle, motorcycle, house, land b) fuel use for cooking, and c) assessment of dowelling characteristics like material of the floor, roof, and house type and elaborated in tables 1,2, and 3. Line 93-96; page 3, 4-5.
Comment:
Of the total sample, how many mothers had a skilled birth attendance?
Response:
There were 204 out of the 810 mothers had a skilled birth attendant. Line 121; page 3
Comment:
In 3.2. I would recommend the author added the results of factors associated with skilled birth attendance.
Response:
We have added the results for skilled birth attendance. Line 134-137; page 5
DISCUSSION
Comment:
The author provides context for the results and compares the findings of this study to other area and other population. Besides mother’s education and socioeconomic status, are they any other potential associated factors?
Response:
Other potential factors associated of health facility utilization in South Sudan and sub-Sahara Africa have been discussed. Line 179-187; page 6.
Comment:
Give some suggestions for next steps that the Sudan government could take to increase access to health facility utilization. It could be discussed more.
Response:
We have suggested the next steps that the South Sudan government could take to increase access to health facility utilization. Line 188-196; page 6.
Comment:
The authors might consider adding some text describing the next steps for research.
Response:
Done. Line 212-215; page 7.
Reviewer 2 Report
The manuscript was written well with minor errors.
Line 46) What do you mean by scaled up?
Line 47) Do you have two intervention or one?
Line 56) Add the year for October
Line 63) Delete the word "the" after 43 villages
Line 85) What does Q2, Q3, and Q4 mean
Lines 68-69) Does it state where the sampling procedures were published?
Line 99) Space after the period
Line 152) Reword for the sentence to clarify..(acquainted how with the facility)
Reference section-Check the spacing and make sure the references are in APA format.

Author Response
We thank you for your detailed and professional suggestions and comments. We are grateful and know that they will make our manuscript professional and easier to read. We have revised the manuscript according to the comments. We have made point by point responses as follows:
Comment:
The manuscript was written well with minor errors.
Response:
Thank you for these kind comment. We have addressed the minor errors in the manuscript
Comment:
Line 46) What do you mean by scaled up?
Response:
We meant interventions must be increased to reduce maternal and neonatal morbidity and mortality. Line 49; page 2.
Comment:
Line 47) Do you have two intervention or one?
Response:
We did not have any intervention. We conducted observational study of two components giving birth in a health facility with the help of skilled birth attendance. Line 49; page 2
Comment:
Line 56) Add the year for October
Response:
Done. Line 58; page 2
Comment:
Line 63) Delete the word "the" after 43 villages
Response:
The word "the" have been deleted. Line 65; page 3.
Comment:
Line 85) What does Q2, Q3, and Q4 mean
Response:
We have defined Q2, Q3 and Q4. Line 92; page 3
Comment:
Lines 68-69) Does it state where the sampling procedures were published?
Response:
The sampling procedures were published in Global Health Action Journal. Line 72-73; page 2
Comment:
Line 99) Space after the period
Comment:
Done. Line 110; page 3 acquainted
Line 152) Reword for the sentence to clarify..(acquainted how with the facility)
Response:
We have re written the sentence. Line 169-171; page 6.
Comment:
Reference section-Check the spacing and make sure the references are in APA format.
Response:
We have checked and corrected the spacing in the reference section.
Round 2
Reviewer 1 Report
OK.